# Cannabis Consumption Used by Cancer Patients during Immunotherapy Correlates with Poor Clinical Outcome

**DOI:** 10.3390/cancers12092447

**Published:** 2020-08-28

**Authors:** Gil Bar-Sela, Idan Cohen, Salvatore Campisi-Pinto, Gil M. Lewitus, Lanuel Oz-Ari, Ayellet Jehassi, Avivit Peer, Ilit Turgeman, Olga Vernicova, Paula Berman, Mira Wollner, Mor Moskovitz, David Meiri

**Affiliations:** 1Cancer Center, Emek Medical Center, 21 Yitzhak Rabin Blvd, Afula 1834111, Israel; idan5161@gmail.com (I.C.); Olga_vr@clalit.org.il (O.V.); 2Bruce Rappaport Faculty of Medicine, Technion-Israel Institute of Technology, Haifa 320002, Israel; Lanuel.fuchs@gmail.com; 3The Laboratory of Cancer Biology and Cannabinoid Research, Department of Biology, Technion-Israel Institute of Technology, Haifa 320003, Israel; campisi.pinto@gmail.com (S.C.-P.); lgil@technion.ac.il (G.M.L.); bermansh@gmail.com (P.B.); 4Statistic unit, Emek Medical Center, Afula 1834111, Israel; ayelletaj@gmail.com; 5Division of Oncology, Rambam Health Care Campus, Haifa 320002, Israel; a_peer@rambam.health.gov.il (A.P.); I_TURGEMAN@rambam.health.gov.il (I.T.); m_wollner@rambam.health.gov.il (M.W.); m_moskovitz@rambam.health.gov.il (M.M.)

**Keywords:** cannabis, endocannabinoids, immune checkpoint inhibitors, cancer

## Abstract

**Simple Summary:**

Cannabis is widely used by patients with cancer to help with cancer symptoms and treatment side effects. Though cannabis has immunomodulatory effects, and its consumption among cancer patients needs to be carefully considered due to its potential effects on the immune system. In this report, we provide the first indication of the impact of cannabis consumption during immune checkpoint inhibitors (ICI) immunotherapy cancer treatment and show it may be associated with worsening clinical outcomes. Cancer patients using cannabis showed a significant decrease in time to tumor progression (TTP) and decreased overall survival (OS) compared to nonusers. In contrast, the use of cannabis reduced immune-related adverse events (iAE). Thus, our report constitutes the first warning sign to the use of cannabis as a palliative treatment in advanced cancer patients starting immunotherapy and suggests that its consumption should be used with attentiveness. Furthermore, we show that the levels of endogenous serum eCB and eCB-like lipids are affected by immunotherapy and may potentially constitute monitoring targets to cancer immunotherapy treatment, which currently has poor clinical markers for predicting patient response rates.

**Abstract:**

Cannabis or its derivatives are widely used by patients with cancer to help with cancer symptoms and treatment side effects. However, cannabis has potent immunomodulatory properties. To determine if cannabis consumption during immunotherapy affects therapy outcomes, we conducted a prospective observatory study including 102 (68 immunotherapy and 34 immunotherapy plus cannabis) consecutive patients with advanced cancers who initiated immunotherapy. Cannabis consumption correlated with a significant decrease in time to tumor progression and overall survival. On the other hand, the use of cannabis reduced therapy-related immune-related adverse events. We also tested the possibility that cannabis may affect the immune system or the tumor microenvironment through the alteration of the endocannabinoid system. We analyzed a panel of serum endocannabinoids (eCBs) and eCB-like lipids, measuring their levels before and after immunotherapy in both groups. Levels of serum eCBs and eCB-like lipids, before immunotherapy, showed no significant differences between cannabis users to nonusers. Nevertheless, the levels of four eCB and eCB-like compounds were associated with patients’ overall survival time. Collectively, cannabis consumption has considerable immunomodulatory effects, and its use among cancer patients needs to be carefully considered due to its potential effects on the immune system, especially during treatment with immunotherapy.

## 1. Introduction

To date, the U.S. Drug Enforcement Administration (DEA) lists cannabis and its phytocannabinoids as Schedule I controlled substances that cannot be legally prescribed, possessed, or sold [1]. Despite the lack of robust evidence, the use of cannabis as a palliative treatment to relieve the side effects of drugs used in a range of medical conditions has been approved or is under consideration in many countries around the world and assumed to be safe [2]. Cannabis is particularly prevalent as a palliative treatment in oncology [2,3,4,5] to alleviate cancer symptoms, including nausea, anorexia, and cancer-related pain [6], despite the limited number of randomized clinical trials. Although the anti-inflammatory effects of cannabis and phytocannabinoids have long been known [7,8], the secondary effect of cannabis on the immune system has not yet been fully elucidated.

The biological activity of cannabis is mediated by modulation of the endocannabinoid system (eCBS). This system comprises of two G-protein-coupled receptors (GPCR), cannabinoid receptors type 1 and 2 (CB1 and CB2, respectively), the two endocannabinoids (eCBs) *N*-arachidonoyl ethanolamide (anandamide, AEA) and 2-arachidonoylglycerol (2-AG), and the enzymes responsible for the biosynthesis and hydrolytic inactivation of these eCBs [9]. More recently, additional receptors, biosynthesizing and degrading enzymes, and eCB-like lipids have been recognized as part of an extended eCBS [10,11,12,13] or an “endocannabinoidome” [11]. The extended eCBS, among other things, regulates immune responses in different cell types, affecting cytokine secretion, the induction of apoptosis, and immune cell activation in innate and adaptive immune responses [14]. The classical eCB, AEA, can produce a dose-dependent inhibition of mitogen-induced T and B human lymphocyte proliferations [15]. On the other hand, 2-AG has both pro- and anti-inflammatory effects. eCB-like lipids like the *N*-acyl ethanolamides: *N*-palmitoyl ethanolamide (PEA), *N*-oleoyl ethanolamide (OEA), and *N*-stearoyl ethanolamide (SEA) have been shown to inhibit both T-cell responses directly [16]. Phytocannabinoids, the natural constituents of cannabis, also interact with the extended eCBS by activating and/or inhibiting eCB and eCB-related receptors, enzymes, and transporters [10,17]. For example, via inhibition of the cellular reuptake of AEA by most neutral phytocannabinoids or cannabidiol (CBD) inhibition of the hydrolyzing enzyme, fatty acid amide hydrolase (FAAH), which, in turn, is responsible for the degradation of AEA and other *N*-acyl ethanolamides [18,19]. Since cannabis is now widely used in oncology patients, it is of great importance to understand the effects of cannabis on the immune system and its possible interaction with immunotherapy, which is designed to artificially stimulate the immune system and improve the immune system’s natural ability to fight the disease.

Several studies show the effect of prolonged use of cannabis on the levels of eCBs. For example, Morgan et al. [20] observed a reduction of AEA levels in the cerebral spinal fluid (CSF), while an increase in 2-AG levels in the serum of chronic cannabis users. Furthermore, Leweke et al. [21] have shown that a two-week treatment with CBD increased circulating levels of AEA, PEA, and OEA in the serum of schizophrenia patients. In a recently published paper by our group [22], we also showed that different high-CBD cannabis extracts modulate the levels of eCB and eCB-like lipids in mouse brains and serum.

Immune checkpoint inhibitors (ICIs) target the molecule’s cytotoxic T-lymphocyte-associated protein 4 (CTLA4), programmed cell death protein 1 (PD-1) (also known as CD279PD-1), and programmed death-ligand 1 (PD-L1) (also known as CD274), either as a monotherapy or in combination with chemotherapy [23,24,25,26,27]. We previously showed in a retrospective observational study that cannabis use significantly reduces the response rates to nivolumab during immunotherapy treatment [28]. In this prospective study, we aimed to evaluate the clinical outcome of cannabis use in patients initiating ICI therapy for advanced malignancies. In parallel, we systematically analyzed a comprehensive panel of serum eCB and eCB-like lipid levels to probe their possible variations as a result of cannabis consumption or immunotherapy treatment to test their potential effect on anticancer treatment or tumor progression. Given the immunomodulatory effects of eCB and eCB-like lipids, we hypothesized in this research that the effect of cannabis use on immunotherapy was possibly in part due to phytocannabinoids’ regulation of the endocannabinoidome. Although the majority of the studies investigated the “classical eCBs” (AEA and 2-AG), other eCB-like lipids have also been shown to regulate the immune system [16], supporting our motivation for analyzing these additional compounds.

## 2. Results

### 2.1. Patient Characteristics

Between 01 September 2016 to 25 September 2018, a total of 102 cancer patients were included in this study. Recruitment included patients with metastatic malignancies (stage IV disease) initiating checkpoint inhibitor treatment: 34 patients used cannabis (cannabis-immunotherapy group: CI-G), and 68 did not use cannabis (immunotherapy group: I-G). About 70% of the patients were male, and more than 50% had non-small cell lung cancer (NSCLC) (Table 1). Some differences between groups were unavoidable: liver metastasis of the immunotherapy group (I-G) (I-G 19%) vs. the immunotherapy-cannabis group (IC-G) (67%, *p* = 0.89), treatment type nivolumab and ipilimumab together (I-G 24% vs. IC-G 12% *p* = 0.15), line of treatment: 76% of patients in the IC-G were given immunotherapy as the second line of treatment vs. 54% in the I-G (odds = 1.40, *p* = 0.03) (Table 1).

Minor differences between groups of users vs. nonusers were found in liver or renal functions or the electrolytes balance before the initiation of the anticancer immunotherapy (Table 2). Differences were also found in terms of the lymphocyte count at baseline (before immunotherapy), where 67% of cannabis users (IC-G group) showed low lymphocyte counts (without leukopenia) vs. 51% in non-cannabis users (I-G group) (one-tailed *p* = 0.08) (Table 2). In other words, the users group had 16% more cases of low lymphocytes at baseline, which corresponds to a low-to-medium difference between groups (Cohen’s h = 0.31).

Among the cannabis users, most of the patients (71%) used the lowest prescribed monthly dose of cannabis (20 grams), and only ten patients used a dose of 30 or 40 grams before and during the immunotherapy treatment period. The use of cannabis had been started nine months to two weeks before the first immunotherapy treatment. The cannabis license in Israel in the years of the study includes only the monthly dose and the type of use, smoked or inhaled (cannabis flowers only), prepared cannabis oil, or combined use. Among the 32 cannabis users, eight used only cannabis oil, and six used combined oil and flowers. The patients had permission to change cannabis products monthly. The data of the different products used by the patients during the study period is limited. 

### 2.2. Overall Response

The overall clinical outcomes were estimated in terms of complete response (CR), partial response (PR), and stable disease (SD) as to the Response Evaluation Criteria in Solid Tumors RECIST 1.1 criteria. Cannabis-users showed a significantly lower percentage of clinical benefit (CR + PR + SD) outcomes: 39% vs. 59% over nonusers (*p* = 0.035). CR and SD indexes were larger over nonusers, although the differences were not statistically significant (CR: I-G = 20% and IC-G = 9%, *p* = 0.1329, SD: I-G = 20% and IC-G = 9%, *p* = 0.13), and PR indexes were equivalent between groups (I-G = 19% and IC-G = 21%, *p* = 0.86). Cannabis users were more likely to show symptoms of progressive disease, namely: *n* = 29 out of *n* = 34 (61%) patients in the IC-G group compared to *n* = 27 patients out of *n* = 68 (41%) in the I-G experienced progressive disease (*p* = 0.035). 

### 2.3. TTP and O.S.

The primary endpoint of the study was the time to tumor progression (TTP) (defined as the time from treatment initiation to time to tumor progression or death from any cause), with overall survival (OS) as a secondary endpoint (defined as the time from treatment initiation to death from any cause). These indexes were compared between cannabis users and nonusers (both groups undergoing immunotherapy treatment). Results show that the median TTP for cannabis-users (IC-G) was 3.4 months (95% CI, 1.8–6.0) vs. 13.1 months (95% CI, 6.0-NA) for nonusers (I-G) (Figure 1). With a minimum follow-up of seven months, the median OS for cannabis use (IC-G) was 6.4 months (95% CI, 3.2–9.7) and 28.5 months (95% CI, 15.6-NA) for nonusers (I-G) (log-rank tests *p* = 0.0025 and *p* = 0.0009, respectively (Figure 2). The analysis was conducted with an adjustment for the line of treatment, and we found a significant estimated hazard ratios of 1.95 (95% CI, 1.17–3.26) for TTP and 2.18 (95% CI, 1.241–3.819) for OS (*p* = 0.011 and *p* = 0.007), respectively. Overall, when compared to cannabis-users, nonusers were associated with more favorable TTP and OS outcomes.

### 2.4. Adverse Events

Adverse events (AEs) were documented during the entire study for both groups, including the most frequent side effects associated with the particular immunotherapy. Overall, cannabis-users reported a lower rate of treatment-related AEs compared with nonusers (Table 3). The most frequent grade 2 and above immune-related AEs were skin toxicity (nine cases (13%) in the I-G vs. two (6%) in the IC-G), thyroid disorders (six cases (9%) of the I-G compared to two (6%) in the IC-G), and colitis (documented in six cases (9%) only in the nonuser group (I-G), *p*-value = 0.094). Additional immune-related AEs such as arthritis, adrenal-insufficiency, and pan-uveitis were reported in single patients and only in the nonuser group. One case of hepatitis was documented in each group. General deterioration and edema were reported in single patients in the cannabis-users group. Although the relation to immunotherapy was not completely defined, they were included as immune-related AEs; a steroids treatment was given, with some improvement. Overall, results indicated a significant reduction in immune-related AEs associated with the use of cannabis during immunotherapy (*p* = 0.057).

### 2.5. eCB and eCB-Like Levels

To further test the possibility that a flow of ectopic Phytocannabinoids in the blood of cannabis users may modulate the eCBS and alter their levels, we analyzed a comprehensive panel of serum eCB and eCB-like lipids and measured their levels in cancer patients before and after immunotherapy from both cannabis users and nonusers. eCB and eCB-like serum concentrations were monitored in blood samples using a novel liquid chromatography-mass spectrometric (LC/MS) method recently developed and validated by our group [22] (Table 4). Four lipids [*N*-eicosapentaenoyl ethanolamide (EPEA), *O*-arachidonoyl ethanolamide (O-AEA), *N*-arachidonoyl alanine (A-Ala), and *N*-arachidonoyl gamma-aminobutyric acid (A-GABA)] resulted below the detection limits in all patients and were excluded from this study. Blood samples taken from 36 patients before starting immunotherapy were analyzed for eCB and eCB-like serum levels: 19 patients from the nonusers group (I-G) and 17 from the cannabis-users group (IC-G) (Table 4). Results indicate that (before immunotherapy) a single eCB-like lipid [i.e., 2-oleoyl glycerol (2-OG)] out of 28 was associated with significantly different levels between groups (*p* < 0.04), while all the other compounds showed no significant differences between groups, suggesting that cannabis consumption had no substantial and/or permanent effects on the observed eCB and eCB-like levels before immunotherapy initiation (Table 4). 

Blood samples of *n* = 14 patients (out of a total of *n* = 36 patients) were reanalyzed after the immunotherapy treatment. We assessed the two-way interaction between immunotherapy (before and after initiation) and medical cannabis (users vs. nonusers) on eCB and eCB-like levels. The levels of 9 out of 28 compounds were significantly affected by the initiation of immunotherapy: Arachidonic acid (AA, *p* < 0.01), 2-AG (*p* < 0.01), *N*-docsatetraenoyl ethanolamide (DtEA, *p* < 0.0001), linolenic acid (LnA, *p* < 0.04), *N*-linoleoyl ethanolamide (LEA, *p* < 0.05), *N*-oleoyl amide (O-Am, *p* < 0.0001), OEA (*p* < 0.05), PEA (*p* < 0.0001), and SEA (*p* < 0.0001) (Figure 3 and Table 4). In particular, the average level of two eCB-like lipids (O-Am and OEA) increased after immunotherapy; otherwise, the average levels dropped sharply after immunotherapy. These trends were equivalent between users and nonusers, thus appearing to be independent of the consumption of medical cannabis. Thus, the data indicate that immunotherapy was associated with several significant variations of eCB and eCB-like levels independently from cannabis consumption.

Next, we tested whether eCB and eCB-like levels at the baseline (before immunotherapy) were likely to be associated with any significant variation of OS (Figure 4**)**. Four lipids correlated with OS time; in particular, SEA, 2-AG, and 2-linoleoyl glycerol (2-LG) showed an inverse association with OS expectations (the higher the concentrations of these compounds, the lower the OS). On the other hand, increasing levels of *N*-arachidonoyl amide (A-Am) were associated with increasing OS expectations (Figure 4). The baseline levels of all the remaining compounds were not associated with significant variations of OS in both MC-user and nonuser groups.

## 3. Discussion

Despite the growing number of cannabis studies showing its efficacy as palliative therapy, its overall effects on the eCBS or the immune systems are mostly overlooked. Nonetheless, the properties of cannabis as a potent anti-inflammatory, immunomodulatory, and immune suppressor are well-known but never before considered in the context of immunotherapy. In this prospective observatory clinical study, we examined the link between cancer immunotherapy and its possible interaction with cannabis. We aimed to better understand the consequences of cannabis consumption during the ICI immunotherapy of patients with advanced cancers. 

Considering that the random assignment of patients simply to "cannabis-users" is highly problematic, we designed a prospective study based on real-life medical records. Primarily, we note that, with this given study design, when differences between treatment groups (cannabis-users vs. nonusers) are inevitable, the effect of cannabis consumption is delimited. We minimized potential biases by making the treatment groups (cannabis-users vs. nonusers) comparable concerning the control variables at enrollment time. In that context, we prioritized the matching of clinical parameters rather than medical treatment history.

In view of the intrinsic limitations of this type of study, and in light of no other prior clinical indications to the subject, our results indicated that cannabis consumption should be carefully considered in patients with advanced malignancies treated with immunotherapy. Our data suggest that exposure to cannabis before or during ICI immunotherapy may associate with worsening success rates. These observations were in-line with our previous retrospective observational study, where cannabis-users were associated with reduced response rates to nivolumab [28]. Indeed, in this current study, our data indicate that cannabis-users were associated with shorter TTP and shorter OS (Figure 1 and Figure 2). Furthermore, lymphocyte counts at the baseline were lower in the cannabis-users group (Table 2), where higher counts correlated positively with the treatment success rate. Remarkably, in the current analysis, cannabis users were also associated with a lower number of immune-mediated iAEs. Of note, it has been shown that patients with advanced melanoma, renal cell carcinoma, or non–small cell lung cancer treated with nivolumab, who developed treatment-related iAEs, were likely to result in significantly prolonged five-year OS [29]. Incidentally, the immunosuppressive effect of cannabis has been highlighted in such a way that clinical tests with selective agonists of cannabinoid receptors (CB1 and CB2) are being evaluated as a new class of immunosuppressive and anti-inflammatory therapeutic targets for autoimmune diseases when the dampening of the immune system is beneficial [7]. 

Cannabinoids and, particularly, various phytocannabinoids have been shown to affect the functional activities of immune cells [7,30,31,32,33,34], while the inhibitory effect of cannabis on lymphocytes has been widely observed before [35]. For example, phytocannabinoids have the potential to modulate the activation and balance of human T-helper 1 (Th1)/T-helper 2 (Th2) cells, lymphocytes, and killer cells [36,37]. Phytocannabinoids and, particularly, (-)-Δ^9^-*trans*-tetrahydrocannabinol (Δ^9^-THC) were found to suppress the ability of T-cells to respond to those mitogen-stimulated [37,38,39]. Importantly, Δ^9^-THC was also shown to differentially suppress CD8 T-cells and cytotoxic T lymphocytes (CTLs) and reduce their cytolytic activity [40] or may trigger T cell exhaustion [41,42]. Additionally, Δ^9^-THC inhibits both the proliferation of lymphocytes responding to an allogeneic stimulus and the maturation of these lymphocytes to mature CTLs [40]. All of the above have the potential to interfere with anticancer biological immunotherapy and were, therefore, our motivation to analyze it in a prospective comparative clinical study currently ongoing in our labs.

We also tested the possibility that cannabis affects the immune system or the tumor microenvironment through the alteration of patients endogenous eCB and eCB-like lipid levels as a result of the addition of exogenous phytocannabinoids (due to cannabis consumption). We systematically analyzed a comprehensive panel of serum eCB and eCB-like lipids measured on cancer patients before and after immunotherapy from both groups. Surprisingly their baseline serum levels, before immunotherapy, showed no significant differences between groups. After immunotherapy, only nine eCB and eCB-like levels showed considerable variations (equally in both groups of cannabis-users and nonusers), while others remained invariant concerning the pre-immunotherapy levels (Figure 3). Collectively, our data imply that eCB and eCB-like levels appear to be affected mostly by immunotherapy rather than by cannabis consumption. Nonetheless, even though phytocannabinoids do not directly alter eCB and eCB-like lipid serum levels, we found an association between the treatment success rate of four of these compounds independently from cannabis exposure (Figure 4), suggesting that the effect of eCB and eCB-like lipids on immunotherapy success rates should be further examined. 

Overall, our current study, with its limitations, demonstrates a statistically significant interaction between cannabis exposure before and during the ICI immunotherapy of advanced malignancies, their lymphocytes counts, and the detected immunotherapy success rates.

## 4. Methods 

### 4.1. Patients 

Patients were recruited consecutively between 1 September 2016 and 25 September 2018. The study took place in the Division of Oncology at Rambam Health Care Campus, Haifa, Israel and was approved by the institutional ethics committee (Certificate 0089-16-RMB). The selection of suitable candidates for the study was conducted through the hospital’s computerized system. Inclusion criteria were patients over 18 years of age and diagnosed stage IV cancer. All patients were recruited during the initiation of checkpoint inhibitor treatment, including anti-PD-1 (Pembrolizumab or Nivolumab), anti-PD-L1 (Durvalumab or Atezolizumab), or combined anti-PD-1 and anti-CTLA4 (Ipilimumab and Nivolumab). All patients gave written informed consent for the prospective evaluation of their medical data and blood samples taken before immunotherapy initiation and after 11–14 weeks of treatment. Data on demographics and medical history was extracted from the medical files. This data included age, type of cancer, stage (according to the American Joint Committee on Cancer, seventh edition), diagnosis date, smoking status, metastases location, treatment line number, immunotherapy details, and cannabis start date. 

### 4.2. Assessments and Analyses

Patients were followed prospectively for tumor responses as evaluated using the RECIST criteria based on imaging assessments carried out every 11–14 weeks. Time to tumor progression (TTP) was calculated as the time from study enrollment until the tumor worsens, spreads, or death (whichever occurred first). Overall survival (OS) was calculated as the time from study enrollment until death from any cause. Only patients who started using cannabis before the initiation of checkpoint inhibitor treatment were included in the cannabis user group. In total, 102 patients were included in this study: 34 patients used cannabis (cannabis-immunotherapy group: CI-G), and 68 did not use cannabis (immunotherapy group: I-G).

### 4.3. Safety

Adverse events (AE) and immune-related AE (iAE) were assessed using the National Cancer Institute Common Terminology Criteria for Adverse Events, version 4.03.32. Adverse events were evaluated throughout the study while participants received treatment and until 90 days after study completion [43]. 

### 4.4. Statistical Analysis

Sample size calculation: We previously reported in a retrospective study that patients with advanced malignancies initiating immunotherapy treatment had a 14% combined (cannabis user and nonuser) rate of complete response (CR) and a 30% rate of partial response (PR) [28]. Accordingly, we calculated that this prospective study required 243 patients (with a group allocation ratio of 1:2, i.e., 81 vs. 162 participants, respectively) to achieve the same difference, with a power of 80% and an alpha of 5% for the two-tailed test. We hypothesized that such difference, with a larger sample size, may translate to significant differences in the TTP. We planned an interim analysis after recruitment of at least 40% of the patients, including a minimum of 30 patients in the cannabis user group, to test our estimation. Patient recruitment was terminated after the interim analysis of 102 patients (*n* = 34 users and *n* = 68 nonusers). A subset of users and nonusers was tested for serum levels before and after therapy initiation. 

A series of $\chi^{2}$ tests or Fisher’s exact tests (when the assumptions of the parametric $\chi^{2}$ test were not met) and nonparametric Mann–Whitney U tests were conducted to analyze the differences between patients’ characteristics in both groups. Time to tumor progression (TTP) and overall survival (OS) was estimated using the Kaplan-Meier survival curve by group, and the log-rank test was computed to differentiate the survival curves between groups. Hazard ratios and the corresponding 95% CIs based on a Cox proportional hazards regression model were provided for multivariate analyses. We computed 2-tailed *p*-values, where *p* < 0.05 was considered a statistically significant result. Statistical analyses were performed using the SAS software package version 9.4 (SAS Institute, Cary, NC, USA).

We used a nonparametric two-way analysis of variance (namely, The Scheirer–Ray–Hare test, SRH) to test the impact of cannabis consumption on immunotherapy, as well as their potential effects on eCB levels. In addition, we used linear regression to summarize the relationship between eCB levels and OS time. Statistical analysis was performed using R statistical software (R Foundation for Statistical Computing, Vienna, Austria) and the dedicated libraries Tidyverse, Tableone, and Stats. 

### 4.5. Measurement of eCB Serum Levels

Analysis of eCB and eCB-like lipids was performed according to a method recently developed and validated by our group [22]. Briefly, liquid chromatography-mass spectrometric (LC/MS) grade acetonitrile, methanol, and water for the mobile phase and high-performance liquid chromatography (HPLC) grade methanol, acetonitrile, and water for sample preparation were obtained from Mercury Scientific and Industrial Products Ltd. (Rosh Haayin, Israel). LC/MS grade acetic acid was purchased from Sigma-Aldrich (Rehovot, Israel). All the standards were of analytical grade (>98%). LnA, AA, EPEA, 2-AG, AEA, and *N*-linoleoyl amide (L-Am) were purchased from Sigma-Aldrich (Rehovot, Israel). All other standards and deuterated internal standards (d-ISs), including: *N*-linolenoyl ethanolamide (LnEA), *N*-docosahexanoyl ethanolamide (DHEA), LEA, PEA, OEA, SEA, DtEA, 2-LG, 2-OG, A-Am, O-Am, 2-arachidonoyl glycerol ether (2-AGe), O-AEA, *N*-arachidonoyl serine (A-Ser), *N*-docosahexaenoyl glycine (DH-Gly), *N*-arachidonoyl glycine (A-Gly), *N*-linoleoyl glycine (L-Gly), *N*-palmitoyl glycine (P-Gly), *N*-oleoyl glycine (O-Gly), A-Ala, *N*-oleoyl alanine (O-Ala) and A-GABA, AA-d8, DHEA-d4, AEA-d4, LEA-d4, PEA-d4, OEA-d4, SEA-d5, 2-AG-d5, A-Ser-d8, and A-Gly-d8, were purchased from Cayman Chemical (Ann Arbor, MI, USA). Six-hundred microliters of the extraction solution (methanol:acetonitrile:acetic acid in a ratio 50:50:0.1) spiked with 20 ng/mL d-ISs were added to 200 µL serum samples. Samples were thoroughly vortexed and centrifuged at 14,000 rpm and 4 ℃ for 20 min to precipitate proteins and cells. The supernatants were then transferred into 3 mL of 0.1% *v*/*v* acetic acid in water and loaded onto preconditioned Agela Cleanert C8 solid-phase extraction (SPE) cartridges (500 mg of sorbent, 50-μm particle size). eCB and eCB-like lipids were eluted from the columns using 2 mL of 0.1% *v*/*v* acetic acid in methanol, evaporated to dryness by speed vac, reconstituted in 100-µl ethanol, and filtered through a 0.22-µm polytetrafluoroethylene (PTFE) syringe filter for LC/MS analysis. Quantification of eCB and eCB-like lipids was performed using a Thermo Scientific ultra-high-performance liquid chromatography (UHPLC) system coupled with a Q Exactive™ Focus Hybrid Quadrupole-Orbitrap MS (Thermo Scientific, Bremen, Germany). The chromatographic separation was achieved using a Halo C18 Fused Core column (2.7 μm, 150 mm × 2.1 mm internal diameter) with a guard column (2.7 μm, 5 mm × 2.1 mm i.d.) (Advanced Materials Technology, Wilmington, DE, USA) and a ternary A/B/C multistep gradient (solvent A: 0.1% acetic acid in Milli Q water, solvent B: 0.1% acetic acid in acetonitrile, and solvent C: methanol; all solvents were of LC/MS grade). The multistep gradient program was established as follows: initial conditions were 50% B raised to 67% B until 3 min, held at 67% B for 5 min, and then increased to 90% B until 12 min, held at 90% B until 15 min, decreased to 50% B over the next min, and held at 50% B until 20 min for re-equilibration of the system prior to the next injection. Solvent C was initially 5% and then lowered to 3% until 3 min, held at 3% until 8 min, raised to 5% until 12 min, and then kept constant at 5% throughout the run. A flow rate of 0.25 mL/min was used, the column temperature was 30 °C, and the injection volume was 1 μL. MS acquisition was carried out with a heated electrospray ionization (HESI-II) ion source operated in switching mode. Source parameters were as follows: sheath gas flow rate, auxiliary gas flow rate, and sweep gas flow rate: 50, 20, and 0 arbitrary units, respectively; capillary temperature: 350 °C; heater temperature: 50 °C; and spray voltage: 3.00 kV. The scan range was 150–750 m/z for all acquisition events. MS was operated in full MS^1^ mode at 70,000 resolution, and the AGC target was set to 10^6^ with a maximum injection time of 100 ms. 

The absolute quantification of eCB and eCB-like lipids was performed by the stable isotope dilution method. Ten-point standard mixes of the analytical standards were prepared in ethanol and spiked with a mixture of all d-ISs at a concentration of 30 ng/mL to yield a calibration range of 0.1 to 1000 ng/mL for all compounds. The ratios of the unlabeled and labeled ions were plotted against the amounts of spiked standards, and the calibration curves were determined empirically according to the weighted least-squares linear regression method, with a weighting factor of 1/X. Since 2-monoacylglycerols (MAGs) spontaneously undergo isomerization to biologically inactive 1-MAGs through the migration of the acyl group from the sn-2 to sn-1/3 position, we summed, in this study, the peaks of 1- and 2-MAGs (total MAGs), as previously suggested in the literature [44,45].

## 5. Conclusions and Study Limitations

The findings of our current prospective observational report recognize the technical limitations of the study design and, so far, lack a mechanism to support the hypothesis presented and should be evaluated in light of the limitations. The study includes a relatively small group of patients in the main clinical categories, such as different cancer types and diverse lines of oncology treatment. Those differences lead to high heterogeneity in the study population. On the other hand, the patients were recruited consecutively, strengthening the daily clinical aspects of the findings, and still, no statistically significant differences were found in the baseline demographic and clinical variables between the study groups, except to the line of treatment that was corrected statistically for the survival analysis. However, other specific characteristics of the tumor, the patient, or the type of immunotherapy treatment made have influences that were not evaluated well due to the sample size. Another significant limitation is the data collected regarding the use of cannabis. All cannabis users included in the study used less than 40 g of cannabis monthly. This amount, according to the Israeli Health Office, is still considered a low amount. However, as mentioned before, in the period of the study, most patients changed cannabis products between months according to cannabis companies recommendations and not medical ones. This aspect of the study prevents homogeneous data regarding the different influences on the immune system that may exist between the various cannabis products.

Nonetheless, we provide here the first indication of the effect of cannabis consumption during ICI immunotherapy treatment and show it may be associated with worsening clinical outcomes when compared to nonusers. Thus, our report constitutes the first warning sign to the use of cannabis as a palliative treatment in advanced cancer patients starting immunotherapy and suggest that its consumption should be carefully considered. 

Additionally, our study shows that, while cannabis consumption does not alter the endogenous levels of serum eCB and eCB-like lipids, their levels appear to be affected by immunotherapy. This may provide monitoring opportunities and potential targets of pharmacological interest to cancer immunotherapy treatment, which currently has poor clinical markers for predicting patient response rates, which remain around 20–35%. Our study provides another demonstration of the fact that cannabis exposure may strongly impact the immune system, and its adverse effects on immunotherapy outcomes should be further tested. It should be noted, however, that eCB and several eCB-like lipids are biosynthesized and metabolized locally and rapidly and are affected by many different factors. Therefore, circulating eCBs often do not represent local disturbances to the eCB tone, and their concentrations are not necessarily related to the microenvironment of the tumor. Additionally, in this study, we could not provide eCB analyses before the cannabis treatment.

## Figures and Tables

**Figure 1 cancers-12-02447-f001:**
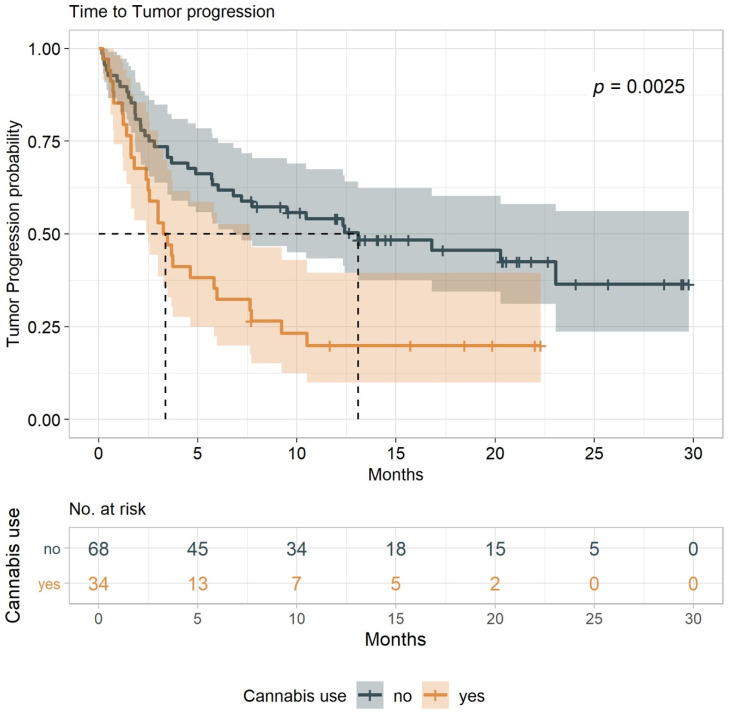
Durability of time to tumor progression (TTP) among patients with advanced cancers receiving immunotherapy, comparing cannabis users to nonusers. Median TTP was 3.4 months (95% confidence interval (CI), 1.8–6.0) for cannabis users and 13.1 months (95% CI, 6.0–NA) for nonusers.

**Figure 2 cancers-12-02447-f002:**
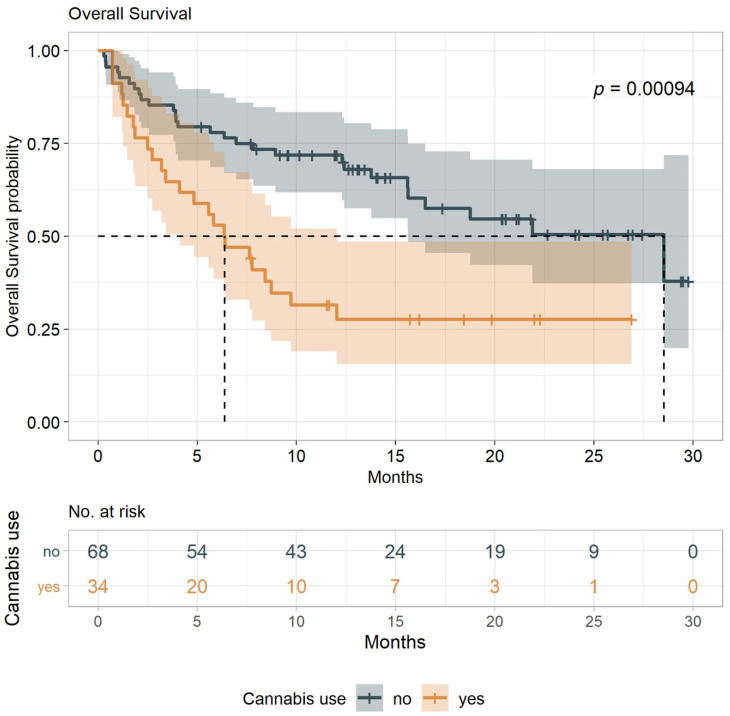
Durability of overall survival (OS) among patients with advanced cancers receiving immunotherapy. Kaplan-Meier estimates of OS among 102 patients divided into cannabis users and nonusers. OS is defined as the time to the last known alive date before the date of data analysis. Median OS was 6.4 months (95% confidence interval (CI), 3.2–9.7) for cannabis users and 28.5 months (95% CI, 15.6–NA) for nonusers.

**Figure 3 cancers-12-02447-f003:**
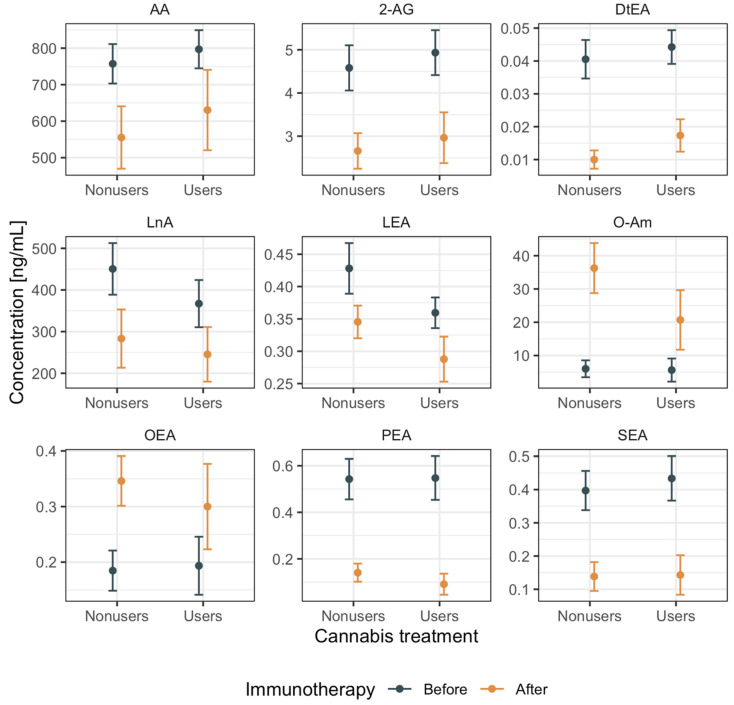
Two-way analysis of endocannabinoid (eCB) and eCB-like lipids variance. Each panel represents a different compound where levels are expressed in (ng/mL) on the y-axis. Dots represent expected values, and lines represent standard error of the mean before and after immunotherapy (grey vs. orange bars) over cannabis users vs. nonusers.

**Figure 4 cancers-12-02447-f004:**
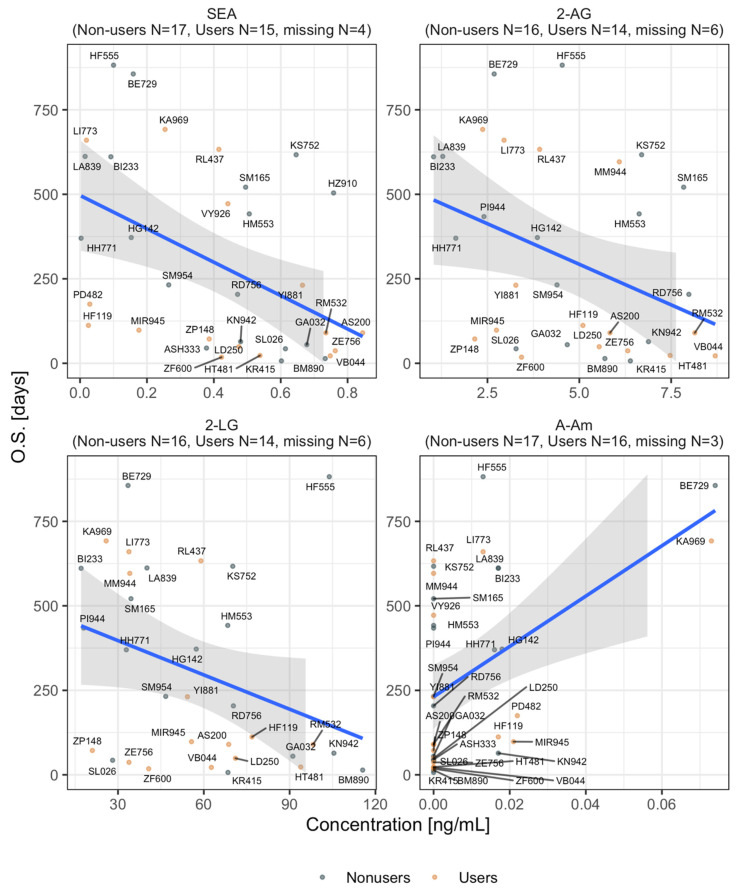
Correlation between eCB concentrations (ng/mL) and overall survival time (OS) measured at time zero (before immunotherapy initiation). Each panel represents a different eCB. Each data point represents a patient with the corresponding anonymized identification number. The blue line indicates the linear trend across the data points. The trends were fitted by means of linear regressions with Gaussian kernel. *N*-arachidonoyl amide (A-Am) includes two extreme values that were considered feasible measurements to be included. The shaded areas indicate a 95% confidence interval of the OS trend. Nonshaded segments indicate that the confidence interval was outside the limits of the y-axis. R_2_/R_2_ adjusted parameters are, respectively: *N*-stearoyl ethanolamide (SEA) = 0.220/0.196, 2-arachidonoylglycerol (2-AG) = 0.139/0.110, 2-linoleoyl glycerol (2-LG) = 0.110/0.080, and A-Am = 0.234/0.211. Estimated *p*-values for the slopes were, respectively: SEA = 0.005, 2-AG = 0.036, 2-LG = 0.064, and A-Am = 0.003. All slopes’ estimate p-values were significant at a 0.1 level, indicating significant variations in OS-time per unit change of patients’ serum eCB concentrations at the baseline.

**Table 1 cancers-12-02447-t001:** Demographics and medical conditions.

Characteristics	Cannabis Non-Users *N* = 68	Cannabis Users *N* = 34	*p*-Value
Age in years median (range)	69 (18–92)	66 (37–85)	
Gender—*N* (%)			
Female	22 (32.4)	10 (29.5)	
Male	46 (67.6)	24 (70.5)	0.9399
Performance Status Eastern Cooperative Oncology Group (ECOG)—*N* (%)			
1≤	55 (80.8)	24 (70.5)	0.3568
≥2	13 (19.1)	10 (29.4)	0.3568
Chronic diseases per patient—*N* (%)			
0	22 (32.3)	13 (22.0)	0.7124
1	16 (23.5)	7 (20.5)	0.9332
2 or more	30 (44.1)	14 (41.1)	0.9437
Background diseases—*N* (%)			
Chronic heart disease	18 (26.4)	5 (14.7)	0.2762
Diabetes	17 (25.0)	6 (17.6)	0.5576
High blood pressure	34 (50.0)	13 (34.1)	0.3612
Chronic obstructive pulmonary disease (COPD)	9 (13.2)	3 (8.8)	1
Hyperlipidemia	23 (33.8)	7 (20.5)	0.2491
Other	2 (2.9)	0 (0.0)	1
Type of malignancy—*N* (%)			
Non-small cell lung cancer	37 (54.4)	20 (58.8)	0.8325
Melanoma	25 (36.7)	9 (26.4)	0.414
Renal cell carcinoma	4 (5.8)	2 (5.8)	1
Other	2 (2.9)	3 (8.8)	1
Main site of metastasis—*N* (%)			
Brain	12 (17.6)	8 (13.2)	0.6593
Lungs	39 (57.3)	23 (67.6)	0.4303
Liver	13 (19.1)	11 (32.3)	0.2157
Immunotherapy given as—*N* (%)			
First line of treatment	31 (45.5)	8 (23.5)	0.05178
Second line of treatment or more	37 (54.4)	26 (76.4)	0.05178
Checkpoint therapy—*N* (%)			
Anti PD1: Pembrolizumab or Nivolumab	47 (69.1)	29 (85.2)	0.127
Ipilimumab and Nivolumab	16 (23.5)	4 (11.7)	0.2517
Anti PDL-1: Durvalumab or Atezolizumab	5 (7.3)	1 (2.9)	1

**Table 2 cancers-12-02447-t002:** Abnormal laboratory tests before immunotherapy (according to local normal ranges).

Test	Cannabis: Nonusers *n* = 68	Cannabis: Users *n* = 34	*p*-Value
Lymphocytes ≤ 1.5 K/uL—N (%)	35 (51)	23 (67)	0.08
Blood count WBC ≤ 4.5 K/uL—N (%)	7 (10)	2 (6)	-
Liver Function			
Alanine Aminotransferase (ALT) > 45	7 (10)	3 (9)	-
Aspartate Aminotrasferase (AST) > 35	8 (12)	5 (15)	-
Alkaline phosphatase level (ALKP) > 120	13 (19)	11 (32)	0.09
Renal Function—N (%)			
Creatinine > 1.17 mL/min	12 (17)	3 (9)	-

**Table 3 cancers-12-02447-t003:** Reported immune-related adverse events (iAE), Common Terminology Criteria for Adverse Events (CTCAE) grade ≥ 2.

Side Effects	Cannabis: Nonusers *n* = 68	Cannabis: Users *n* = 34	*p*-Value
Any iAE, grade ≥ 2—N (%)	28 (39)	7 (21)	0.057
Skin toxicity—N (%)	9 (13)	2 (6)	-
Colitis—N (%)	6 (9)	0	-
Thyroid disorders—N (%)	6 (9)	2 (6)	-
Other—N (%)	3 (4)	0	-
Arthritis—N (%)	1 (1.5)	0	-
Panuveitis—N (%)	1 (1.5)	0	-
Hepatitis—N (%)	1 (1.5)	1 (3)	-
General Deterioration—N (%)	0	1 (3)	-
Edema—N (%)	0	1 (3)	-

**Table 4 cancers-12-02447-t004:** Endocannabinoids serum concentration.

Analyte (ng/mL)	before Immunotherapy Cannabis Nonusers	before Immunotherapy Cannabis Users	after Immunotherapy Cannabis Nonusers	after Immunotherapy Cannabis Users	SMD
	** *n = 8* **	** *n = 6* **	** *n = 8* **	** *n = 6* **	
	Median (IQR)	Median (IQR)	Median (IQR)	Median (IQR)	
LnA	432.52 (211.53, 522.96)	240.99 (228.69, 298.71)	184.35 (155.67, 365.81)	252.72 (118.91, 312.50)	0.443
AA	685.94 (449.35, 831.05)	627.04 (563.61, 700.79)	457.61 (412.89, 657.92)	564.63 (549.41, 583.55)	0.315
EPEA	0.00 (0.00, 0.00)	0.00 (0.00, 0.03)	0.00 (0.00, 0.00)	0.00 (0.00, 0.00)	-
LnEA	0.00 (0.00, 0.03)	0.00 (0.00, 0.00)	0.00 (0.00, 0.01)	0.01 (0.00, 0.03)	0.351
DHEA	0.34 (0.29, 0.40)	0.33 (0.28, 0.37)	0.29 (0.27, 0.34)	0.25 (0.22, 0.29)	0.41
AEA	0.31 (0.26, 0.41)	0.35 (0.29, 0.43)	0.32 (0.25, 0.35)	0.25 (0.20, 0.34)	0.559
LEA	0.36 (0.35, 0.48)	0.41 (0.33, 0.45)	0.36 (0.31, 0.39)	0.28 (0.24, 0.33)	0.745
PEA	0.17 (0.11, 0.21)	0.08 (0.04, 0.21)	0.11 (0.06, 0.21)	0.04 (0.01, 0.18)	0.45
OEA	0.35 (0.27, 0.38)	0.35 (0.31, 0.46)	0.31 (0.26, 0.39)	0.25 (0.19, 0.30)	0.412
SEA	0.13 (0.07, 0.24)	0.10 (0.03, 0.23)	0.10 (0.05, 0.17)	0.11 (0.02, 0.24)	0.335
DtEA	0.01 (0.01, 0.02)	0.02 (0.02, 0.03)	0.01 (0.00, 0.02)	0.02 (0.01, 0.03)	0.582
2-AG	3.27 (1.55, 5.12)	4.02 (2.80, 5.84)	2.18 (1.78, 3.78)	3.06 (2.70, 4.58)	0.472
2-LG	48.74 (33.36, 104.28)	44.91 (33.91, 71.57)	50.95 (35.25, 73.29)	73.83 (38.35, 100.75)	0.352
2-OG	71.84 (36.88, 134.98)	52.58 (38.89, 93.19)	71.59 (52.16, 95.28)	77.85 (57.94, 84.19)	0.338
A-Am	0.02 (0.02, 0.03)	0.02 (0.01, 0.02)	0.02 (0.01, 0.03)	0.01 (0.00, 0.01)	0.607
L-Am	4.65 (1.81, 15.28)	44.44 (15.32, 55.42)	52.71 (10.04, 63.00)	30.45 (9.10, 43.01)	0.625
O-Am	12.85 (7.11, 15.89)	6.68 (3.19, 19.69)	35.98 (19.39, 58.34)	13.85 (6.39, 23.43)	0.384
2-AGe	0.49 (0.46, 0.51)	0.41 (0.40, 0.45)	0.45 (0.41, 0.48)	0.44 (0.39, 0.50)	0.478
O-AEA	0.00 (0.00, 0.00)	0.00 (0.00, 0.00)	0.00 (0.00, 0.00)	0.00 (0.00, 0.00)	-
A-serine	0.22 (0.00, 0.23)	0.22 (0.22, 0.23)	0.23 (0.22, 0.23)	0.22 (0.22, 0.23)	0.647
DH-Gly	0.00 (0.00, 0.42)	0.00 (0.00, 0.00)	0.20 (0.00, 0.41)	0.00 (0.00, 0.00)	0.422
A-Gly	0.07 (0.00, 0.15)	0.07 (0.00, 0.14)	0.07 (0.00, 0.14)	0.00 (0.00, 0.10)	0.234
L-Gly	0.37 (0.33, 0.43)	0.26 (0.24, 0.31)	0.32 (0.28, 0.38)	0.26 (0.19, 0.37)	0.606
P-Gly	0.92 (0.82, 1.11)	0.90 (0.80, 1.15)	0.90 (0.79, 1.04)	0.78 (0.66, 1.04)	0.352
O-Gly	0.64 (0.49, 0.68)	0.49 (0.40, 0.65)	0.54 (0.41, 0.60)	0.31 (0.28, 0.51)	0.524
A-Ala	0.00 (0.00, 0.15)	0.00 (0.00, 0.00)	0.00 (0.00, 0.00)	0.00 (0.00, 0.00)	-
O-Ala	0.40 (0.35, 0.49)	0.36 (0.35, 0.36)	0.39 (0.31, 0.44)	0.34 (0.34, 0.37)	0.586
A-GABA	0.00 (0.00, 0.00)	0.00 (0.00, 0.00)	0.00 (0.00, 0.00)	0.00 (0.00, 0.00)	-

Main descriptive statistics by groups, data = Median, interquartile range (IQR); SMD, standardized mean deviation between groups; LnA, linolenic acid; AA, arachidonic acid; EPEA, *N*-eicosapentaenoyl ethanolamide; LnEA, *N*-linolenoyl ethanolamide; DHEA, *N*-docosahexanoyl ethanolamide; AEA, *N*-arachidonoyl ethanolamide; LEA, *N*-linoleoyl ethanolamide; PEA, *N*-palmitoyl ethanolamide; OEA, *N*-oleoyl ethanolamide; SEA, *N*-stearoyl ethanolamide; DtEA, *N*-docsatetraenoyl ethanolamide; 2-AG, 2-arachidonoyl glycerol; 2-LG, 2-linoleoyl glycerol; 2-OG, 2-oleoyl glycerol; A-Am, *N*-arachidonoyl amide; L-Am, *N*-linoleoyl amide; O-Am, *N*-oleoyl amide; 2-AGe, 2-arachidonoyl glycerol ether; O-AEA, *O*-arachidonoyl ethanolamide; A-Ser, *N*-arachidonoyl serine; DH-Gly, *N*-docosahexaenoyl glycine; A-Gly, *N*-arachidonoyl glycine; L-Gly, *N*-linoleoyl glycine; P-Gly, *N*-palmitoyl glycine; O-Gly, *N*-oleoyl glycine; A-Ala, *N*-arachidonoyl alanine; O-Ala, *N*-oleoyl alanine; A-GABA, *N*-arachidonoyl gamma-aminobutyric acid.

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
