# Peer review of "Cannabis Consumption Used by Cancer Patients during Immunotherapy Correlates with Poor Clinical Outcome"

_cancers, 2020, doi:10.3390/cancers12092447_

Round 1
Reviewer 1 Report
In this study Bar-Sela et al analyse in a prospective study whether chronic cannabis consumption may interfere with the efficacy of immuno-therapy check point inhibitors in cancer patients. Whereas the idea of the study is of great interest and potential relevance, as acknowledged by the authors the work has several important limitations.
Major general concerns:
- Perhaps one of the main limitations of the study is the heterogeneity of the patient population. Thus, the study analyses data derived from patients with lung cancer, melanoma and renal cell carcinoma some of them with metastasis and some not and some of them receiving ICIs as first line or as second line (or more) therapy. Likewise, the distribution of the patients in cannabis non-users and cannabis users is problematic as there might be very important differences in the doses and the duration of the treatment that each patient received. Thus, irrespectively of the existence of statistical differences between some of the parameters analysed, in the absence of additional patients or studies allowing a more homogeneous distribution of the patients, it is not possible to know whether the observed differences can be attributed to the treatment with cannabis extracts or to other specific characteristic of the tumor, the patient, the type of ICI treatment etc. I am aware that these limitations are imposed by the type of study and that the authors tried to be as homogeneous and consistent as they could with the number of patients and information that they were able to gather. However, although some of the observations are of potential interest, in my opinion this experimental design limits very significantly the soundness and relevance of the study.
- Another concern has to do with the lack of mechanistic evidence for the existence of a negative influence of cannabis exposure to ICIs effects. Although ICIs have revolutionized treatments for some tumor types, the percentage of patients that respond to immunotherapy remains in around 20-35%. Thus, 65-80% of the patients subjected to these treatments do not respond at all whereas a certain percentage of the patients become long-term survivors. The reasons for this situation are not clear and may have to do with the mutational load, the influence that the suppression of the immune response has in the origin of the tumor etc. In this context, it would be extremely interesting to know whether treatment with THC or other cannabinoids influence the effectiveness of treatments with ICIs. However, to this aim - among many other parameters – it would be important to know whether the percentage of patients that responds to ICIs (long-term survivors) in each cancer type is modified by the treatment with cannabinoids. Otherwise and in the absence of mechanistic studies that support how treatment with these agents affect the reactivation of the immune response upon treatment with ICIs in these patients, these observations might be biased and therefore may not reflect the effect of cannabinoids on the efficacy of the ICIs.
Other points
- The part of the study relative to the levels of endocannabinoids in the patients is not completely well justified. Why would the authors expect that treatment with phytocannabinoids should influence endocannabinoid levels in the patients? And more important what would be the rational to connect this effect with the efficacy of ICIs in those patients?
- The authors determine the levels of different lipids including various acyl-ethanolamines. However, being strict most of these lipids do not bind with high affinity to cannabinoid receptors and therefore cannot be considered canonical endocannabinoids
- Some of the figures are incomplete/are lacking part of a panel
Author Response
Reviewer-1
Heterogeneity of the patient population and the distribution of the patients in cannabis non-users and cannabis users is problematic as there might be significant differences in the doses and the duration of the treatment that each patient received-
Although in principle, we agree with the reviewer remarks, our findings in this current important study should indeed be evaluated in light of the study's limitations. Even though, it is essential to note that this is a prospective observational study that included a relatively small group of patients in main clinical categories, such as different cancer types and diverse lines of oncology treatment which patients were recruited consecutively Paragraph: strengthen the daily clinical aspects of the finding, and still, no statistically significant differences were found in baseline demographic and clinical variables between the study groups, except to the line of treatment that was corrected statistically for the survival analysis despite the heterogeneity in the study population.
Yet, in our view, we firmly believe that it provides the first indication of the effect of cannabis consumption during ICI immunotherapy treatment and shows Paragraph: it may be associated with worsening clinical outcomes when compared to non-users. Thus, it imperative clinical massage constitutes the first warning sign to the use of cannabis as a palliative treatment in advanced cancer patients starting immunotherapy and suggests that its consumption should be carefully considered.
Therefore, as also suggested by other reviewers, we now dedicated a paragraph, after the discussion, to detail and highlight study limitations. To better stress the overall message while presenting and acknowledging its limitations we also integrated a new, and more moderated, massage to the readers which better consider report limitations and reflect reviewers valuable recommendations (please see Conclusions and study limitations)
Regarding the concern about the study experimental design, we would like to note that this is an observational study. According to the definition (see citation): Observational study design is about the possible effect of a treatment on subjects, where the assignment of subjects into treatment groups is outside the control of the investigator for ethical or logistic reasons. In our case, assignment to cannabis treatment is outside our control because of ethical concerns (please see footnote).
Porta M, ed. (2008). A Dictionary of Epidemiology (5th ed.). New York: Oxford University Press. http://www.academia.dk/BiologiskAntropologi/Epidemiologi/PDF/Dictionary_of_Epidemiology__5th_Ed.pdf
Lack of mechanistic evidence- Surely observational studies cannot be used to make definitive statements of fact about the “safety, efficacy, or effectiveness” of a practice, nor the “mechanism of action of treatment”. With this observational study we can provide information on the “real world” use and practice of cannabis in conjunction with immunotherapy; detect “signals” about the benefits and risks of the of methods in the general population; help formulate hypotheses to be tested in subsequent experiments; provide part of the community-level data needed to design more informative pragmatic clinical trials, and inform clinical practice.
Indeed, we do agree with the reviewer that "It would be extremely interesting to know whether treatment with THC or other cannabinoids influences the effectiveness of treatments with ICIs" and ongoing projects in our labs aim to characterize T cells and or MDCS populations are currently being performed. Consequently, we then added a clear statement of this in the discussion " All of the above have the potential to interfere with anti-cancer biological immunotherapy and were, therefore, our motivation to analyze it in a prospective, comparative clinical study currently ongoing in our labs."
Reviewer: Some of the figures are incomplete/are lacking part of a panel- Figure-3 and Figure-4 are now modified as suggested, and Tables-1 and Table-4 were supplemented with additional data.
Reviewer: Why would the authors expect that treatment with phytocannabinoids should influence endocannabinoid levels in the patients? And more importantly, what would be rational to connect this effect with the efficacy of ICIs in those patients? And also, Reviwer: the authors determine the levels of different lipids, including various acyl-ethanolamine. However, being strict most of these lipids do not bind with high affinity to cannabinoid receptors and therefore cannot be considered canonical endocannabinoids. These issues are now better elaborated and addressed explicitly in the introduction, including many examples and supporting citations:
Paragraph: More recently, additional receptors, biosynthesizing and degrading enzymes, and eCB-like lipids have been recognized as part of an extended eCBS [10–13] or an "endocannabinoidome" [11]. The extended eCBS, among other things, regulates immune responses in different cell types, affecting cytokine secretion, induction of apoptosis, and immune cell activation in innate and adaptive immune responses [14]. The classical eCB, AEA, can produce dose-dependent inhibition of mitogen-induced T and B human lymphocyte proliferation [15]. On the other hand, 2-AG has both pro‐and anti-inflammatory effects. eCB-like lipids like the N-acyl ethanolamides, N-palmitoyl ethanolamide (PEA), N-oleoyl ethanolamide (OEA), and N-stearoyl ethanolamide (SEA), have been shown to inhibit both T-cell responses directly [16]. Phytocannabinoids, the natural constituents of cannabis, also interact with the extended eCBS by activating and/or inhibiting eCB and eCB-related receptors, enzymes, and transporters [10,17]. For example, via inhibition of the cellular reuptake of AEA by most neutral phytocannabinoids, or cannabidiol (CBD) inhibition of the hydrolyzing enzyme, fatty acid amide hydrolase (FAAH), which in turn is responsible for the degradation of AEA and other N-acyl ethanolamides [18,19]. Since cannabis is now widely used in oncology patients, it is of great importance to understand the effects of cannabis on the immune system and its possible interaction with immunotherapy, which is designed to artificially stimulate the immune system and improve the immune system natural ability to fight the disease.
Paragraph: Several studies show the effect of prolonged use of cannabis on the levels of eCBs. For example, Morgan et al. [20] have observed a reduction of AEA levels in the cerebral spinal fluid (CSF), while an increase in 2-AG levels in the serum of chronic cannabis users. Furthermore, Leweke et al. [21] have shown that a two-week treatment with CBD increased circulating levels of AEA, PEA, and OEA in the serum of schizophrenia patients. In a recently published paper by our group [22], we also show that different high-CBD cannabis extracts modulate the levels of eCB and eCB-like lipids in mouse brain and serum.
Taken together with the new section highlighting the study limitations, we added a new, more moderated concluding massage to the readers, which better consider and reflect the report limitations raised by the reviewer.
Paragraph: "We provide here the first indication of the effect of cannabis consumption during ICI immunotherapy treatment and show it may be associated with worsening clinical outcomes when compared to non-users. Thus, our report constitutes the first warning sign to the use of cannabis as a palliative treatment in advanced cancer patients starting immunotherapy and suggests that its consumption should be carefully considered."
Reviewer 2 Report
The prospective work performed by Bar-Sela G. and colleagues entitled “Chronic Cannabis consumption used by cancer patients during immunotherapy correlates with poor clinical outcome” describes the impact and consequences of cannabis use in patients with advance cancers under ICI immunotherapy.
In my opinion, the study is very interesting and of a major importance and value, as the topic of cannabis consumption is very debated nowadays. In accordance with the authors, although there are many studies presenting the efficacy of cannabis use in palliative therapy, the impact of cannabis consumption in the context of immunotherapy has never been shown.
I find this study being well conceived (designed). The results are very well described and presented and the article is well written. It follows a good logic and the discussion part stays on the topic.
However, I have some suggestions that might improve the manuscript:
- Some phrases of the manuscript require English editing (e.g.: lines 27 – 29 and others)
- Study limitations – could be better described
- And also further perspectives should be stated
- For lines 270 and 280 – please include citation
It should have been interesting to also see differences between cannabis preparations in terms of clinical outcome.
Author Response
Reviewer-2
Some phrases of the manuscript require English editing (e.g., lines 27 – 29 and others). Lines 27-29 are now rephrased, and an English professional editor edited the manuscript
Study limitations – could be better described and also, further perspectives should be stated. As also suggested by other reviewers, we now allocated a special dedicated section after the discussion to detail and highlight study limitations in a special section. To better stress, the overall message while concurrently present and acknowledge its limitations together we also integrated a new more moderated perspective massage to the readers which better consider and reflect the report limitations (please see Conclusions and Study limitations)
For lines 270 and 280 – please include citation. Citation in these lines is now included and incorporated
Reviewer 3 Report
The manuscript “Chronic Cannabis consumption used by Cancer patients during immunotherapy correlates with poor clinical outcome” is an interesting study that describes decreased survival rates for cancer patients who used Cannabis during immunotherapy. There is a huge amount of data describing that phytocannabinoids and also endocannabinoids may regulate immune cell functions so the rationale was clear to look into effects of Cannabis during immunotherapy. Patient data are urgently needed in this field. The role of the bioactive lipids measured in the study is not so clear as many of them do not bind to cannabinoid receptors themselves but are rather responsible for the entourage effect. The immunomodulatory link of the eCBs measured in the study has no solid rationale and needs more explanantion. Authors need to add a more informative limitation section in which they describe and discuss the raised concerns.
Some major concerns are outlined below
Patient characteristics: a better characterization of the Cannabis user group would have been desirable. Are there data how the participants used Cannabis (oil? inhaled? how often?). This is a weak point as Cannabis effects are very much concentration dependent.
“We also tested the possibility that cannabis may affect the immune system or the tumor microenvironment through the alteration of the endocannabinoids system”
What was the rationale to test bioactive lipids? Although claimed in the abstract that “immunomodulatory effects” of Cannabis are tested, bioactive lipids were measured instead which does not inform about immunomodulatory effects. To test the immune system authors probably need to use flow cytometry and look at the leukocyte population of blood/tumor. These data are not presented. Only total lymphocyte counts are presented.
Please, do not catagorize all lipids measured in the study as eCBs, this is confusing and incorrect. Secondly, the abbreviations of the lipids in Table 4 need to be stated in the legend. Maybe authors can catagorize the lipids as endocannabinoids (AEA, 2-AG), and endocannabinoid-related or -like lipids (which are mosty N-acylethanolamines and their derivatives). Although some of the lipids corellate with OS that does by no means imply that this is tumor microenvironment-derived. eCB production is widespread in the body and rather act locally.
A big caveat with measuring endocannabinoids is that the production of these lipids still continues in the blood shortly after collection (see Gurke et al. Talanta. 2019;204:386-394. doi:10.1016/j.talanta.2019.06.004), this means that one needs to employ strict preanalytics (spinning down immediately and quickly frozen) to get meaningful data.There is no mention on preanalytics in the Methods section. A statement in the limitations dealing with this concern is strongly recommended.
In the Abstract it is stated: serum endocannabinoids (line 25) while in Tabe 4 it is stated “Plasma endocannabinoid level” (line 157). What was used for lipid measurement, serum or plasma? Please correct.
line 152: MC users group. You mean medical cannabis?
Table 1: PS is an abbreviation for what?
line 52: cannabimimetic lipids is a somewhat incorrect expression, you probably mean enocannabinoid-like lipid?
How were the correlations calculated? Please indicate the r2 and p values for the regression
Figure 4: What do the numbers in the graphs stand for?
Author Response
Reviewer-3
The immunomodulatory link of the eCBs measured in the study has no solid rationale and needs more explanation. We also tested the possibility that cannabis may affect the immune system or the tumor microenvironment through the alteration of the endocannabinoid system? And what was the rationale to test bioactive lipids? These issues are now better elaborated and addressed explicitly in the introduction, including many examples and supporting citations:
Paragraph: More recently, additional receptors, biosynthesizing and degrading enzymes, and eCB-like lipids have been recognized as part of an extended eCBS [10–13] or an "endocannabinoidome" [11]. The extended eCBS, among other things, regulates immune responses in different cell types, affecting cytokine secretion, induction of apoptosis, and immune cell activation in innate and adaptive immune responses [14]. The classical eCB, AEA, can produce dose-dependent inhibition of mitogen-induced T and B human lymphocyte proliferation [15]. On the other hand, 2-AG has both pro‐and anti-inflammatory effects. eCB-like lipids like the N-acyl ethanolamides, N-palmitoyl ethanolamide (PEA), N-oleoyl ethanolamide (OEA), and N-stearoyl ethanolamide (SEA), have been shown to inhibit both T-cell responses directly [16]. Phytocannabinoids, the natural constituents of cannabis, also interact with the extended eCBS by activating and/or inhibiting eCB and eCB-related receptors, enzymes, and transporters [10,17]. For example, via inhibition of the cellular reuptake of AEA by most neutral phytocannabinoids, or cannabidiol (CBD) inhibition of the hydrolyzing enzyme, fatty acid amide hydrolase (FAAH), which in turn is responsible for the degradation of AEA and other N-acyl ethanolamides [18,19]. Since cannabis is now widely used in oncology patients, it is of great importance to understand the effects of cannabis on the immune system and its possible interaction with immunotherapy, which is designed to artificially stimulate the immune system and improve the immune system natural ability to fight the disease.
Several studies show the effect of prolonged use of cannabis on the levels of eCBs. For example, Morgan et al. [20] have observed a reduction of AEA levels in the cerebral spinal fluid (CSF), while an increase in 2-AG levels in the serum of chronic cannabis users. Furthermore, Leweke et al. [21] have shown that a two-week treatment with CBD increased circulating levels of AEA, PEA, and OEA in the serum of schizophrenia patients. In a recently published paper by our group [22], we also show that different high-CBD cannabis extracts modulate the levels of eCB and eCB-like lipids in mouse brain and serum.
The authors need to add a more informative limitation section in which they describe and discuss the raised concerns. As also suggested by other reviewers, we now allocated a special dedicated section after the discussion to detail and highlight study limitations in a special section. To better stress the overall message while concurrently present and acknowledge its limitations together, we also integrated a new, more moderated perspective massage to the readers, which better consider and reflect the report limitations (please see Conclusions and Study limitations).
A better characterization of the Cannabis user group would have been desirable. Are there data on how the participants used cannabis (oil? inhaled? how often?). We now added the following information to the results section under Patient Characteristics: Among the cannabis users, most of the patients (71%) used the lowest prescribed monthly dose of cannabis (20 grams), and only ten patients used a dose of 30 or 40 grams before and during the immunotherapy treatment period. The use of cannabis had been started nine months to 2 weeks before the first immunotherapy treatment. The cannabis license in Israel in the years of the study includes only the monthly dose and the type of use, smoked or inhaled (cannabis flowers only), prepared cannabis oil, or combined use. Among the 32 cannabis users, 8 used only cannabis oil, and 6 used combined oil and flowers. The patients had permission to change cannabis products monthly. The data of the different products used by the patients during the study period is limited.
To test the immune system, authors probably need to use flow cytometry and look at the leukocyte population of blood/tumors. These data are not presented. Only total lymphocyte counts are presented. Indeed, we share the same view with the reviewer when ongoing projects in our labs aim to characterize T cells and or MDCS populations are currently being performed. We believe that a full characterization of such mechanism is beyond the scope of this observational study, which cannot be used to make definitive statements of fact about the "safety, efficacy, or effectiveness" of a practice, nor the "mechanism of action of treatment. This observational study aimed to provide information on the "real world" use and practice, detect signals about the benefits and risks of...[the] use [of methods] in the general population; help formulate hypotheses to be tested in subsequent experiments, provide part of the community-level data needed to design more informative pragmatic clinical trials, and inform clinical practice.
Consequently, we then added a clear statement of this in the discussion " All of the above have the potential to interfere with anti-cancer biological immunotherapy and were, therefore, our motivation to analyze it in a prospective, comparative clinical study currently ongoing in our labs."
The abbreviations of the lipids in Table 4 need to be stated in the legend
Table-4 is now modified as suggested
There is no mention on preanalytics in the Methods section. A statement in the limitations of dealing with this concern is strongly recommended.
In the Abstract, it is stated: serum endocannabinoids (line 25), while in Table-4, it is stated: "Plasma endocannabinoid level" (line 157). What was used for lipid measurement, serum, or plasma? Please correct. The text in Table-4 was modified to serum.
Line 152: was corrected to cannabis, PS (as for Performance Status) in Table-1 is an abbreviation that was added as requested. Line 52: modified to endocannabinoid-like lipid as suggested.
Reviewer 4 Report
The paper is addressing a very important topic regarding the use of cannabinoids during the treatment (immunotherapy) of cancer patients.
The paper must be carefully revised in aim to present the data in adequate way. Major concerns are listed below, additional comments are indicated it attached file:
- The number of patients per group (I, IC) does not seems to be adequate (table 1).
- Indicate the disease stage and do include it in the presentation of the TTP and OS (figure 1 , 2).
- Only the same patients before & after the ImTh must presented (table 4)

Author Response
The number of patients per group (I, IC) does not seem to be adequate (Table 1). Table-1 is now modified and contains a better and elaborated presentation of the data. We also added a few more sections or remarks to enhance the clarity of the presented items.
Indicate the disease stage and do include it in the presentation of the TTP and OS (figure 1, 2). All Recruitment included patients with metastatic malignancies (stage IV disease). This is now clearly stated in the text, and the following text was added to the Results section under Patient Characteristics.
Recruitment included patients with metastatic malignancies (stage IV disease) initiating checkpoint inhibitor treatment: 34 patients used cannabis (Cannabis-Immunotherapy group: CI-G), and 68 did not use cannabis (Immunotherapy group: I-G). About 70% of the patients were male, and more than 50% had Non-Small Cell Lung Cancer (NSCLC) (Table-1).
Only results from the same patients (N = 8 and N = 6) have been included in table 4- Modified as requested. Line 173: Figure 3, changed accordingly.
Line 184: On figure 4, Am-m concentrations for two patients were extreme, but we decided to include them because they were valid measurements.
On figure 4- N were added, Line 284: 1:2 Ratio defined
The text changes on the attached PDF files were all incorporated according to the reviewer's suggestions, and the Abbreviation list was added as suggested.
Round 2
Reviewer 1 Report
I appreciate the effort that the authors have made in preparing a revised version of the manuscript where they have expanded the introduction, added additional explanations and underlined the limitations of the study. Nonetheless, in my opinion the major concerns and limitations of the study identified in my previous report remain in the revised version of the manuscript.
Author Response
We have now incorporated the statement suggested by you and Reviewers -1 in the new version under Conclusions and Study limitations.
Now the opening message is as follows:
"The findings of our current prospective observational report recognize the technical limitations of the study design, and so far, lack a mechanism to support the hypothesis presented and should be evaluated in light of the limitations"
Reviewer 3 Report
There are no further concerns
Author Response

(The authors gave the same response as above.)
